# A retrospective cohort study of major adverse cardiac events in children affected by Kawasaki disease with coronary artery aneurysms in Thailand

**Kanokvalee Santimahakullert[1], Chodchanok Vijarnsorn[1]\*, Yuttapong Wongswadiwat[2], Prakul Chanthong[1], Sappaya Khrongsrattha[1], Manat Panamonta[2], Paradorn Chan-on[2], Kritvikrom Durongpisitkul[1], Paweena Chungsomprasong[1], Supaluck Kanjanauthai[1], Jarupim Soongswang[1]**

1 Division of Pediatric Cardiology, Department of Pediatrics, Faculty of Medicine, Siriraj Hospital, Mahidol University, Bangkok, Thailand, 2 Division of Pediatric Cardiology, Department of Pediatrics, Faculty of Medicine, Srinagarind Hospital, Khon Kaen University, Khon Kaen, Thailand

\* cvijarnsorn@yahoo.com

**Data Availability Statement:** The datasets generated and/or analyzed during the current study

## Abstract

Kawasaki disease (KD) is a common form of vasculitis in children that can be complicated by coronary artery aneurysms (CAAs). Data of long-term outcomes and major adverse cardiac events (MACE) in children with CAAs following KD in developing country are limited. Our aims were to determine the rates of MACE and identify risk factors associated with MACE in children with KD and CAAs in Thailand. We performed a retrospective analysis of data from 170 children diagnosed with KD and CAAs in two tertiary hospitals between 1994 and 2019. During a median (range) follow-up of 5.4 years (22 days to 23 years), 19 patients (11.2%) experienced MACE, that included 12 coronary artery bypass grafting, 2 percutaneous coronary intervention and 5 children with evidence of myocardial ischemia and coronary occlusion. Coronary interventions were performed at a median time of 4 years (0.01 to 9.5 years) after KD diagnosis. Forty-nine patients (28.8%) had giant CAAs. No MACE was reported in children with small CAAs. Independent risks of MACE were from the absence of intravenous immunoglobulin treatment (HR 7.22; 95% CI 2.21 to 23.59; $p = 0.001$), the presence of giant aneurysms (HR 13.59; 95% CI 2.43 to 76.09; $p = 0.003$), and CAAs that involved bilateral branches of coronary arteries (HR 6.19; 95% CI 1.24 to 30.92; $p = 0.026$). Among children with giant CAAs, the intervention-free rate was 93.8%, 78.7% and 52.2%, at 1, 5 and 10 years, respectively. Of note, 81% of the small CAAs regressed to a normal size, and for medium CAAs, 50% regressed to normal size. Overall, ~10% of children with CAAs following KD experienced MACE in this cohort. Timely IVIG treatment in children with KD following symptom onset will reduce the risk of MACE. Cautious surveillance to identify cardiac complications should be recommended for children once medium or giant CAAs develop.

**Trial registration:** TCTR20190125004.

are not publicly available due to patient
confidentiality, but they are available from the
corresponding author and ethics committee
(contact via siriraj_irb@mahidol.ac.th) upon
reasonable request.

**Funding:** The authors received no specific funding
for this work.

**Competing interests:** The authors have declared
that no competing interest exist.

## Introduction

Kawasaki disease (KD) is an acute febrile illness involving inflammation of medium-sized vessels, which primarily affects young children [1, 2]. Without treatment, patients can develop complications, including coronary artery aneurysms (CAAs) in 15% to 25% of cases [2, 3]. Intravenous immunoglobulin (IVIG) treatment within 10 days of symptom onset leads to a reduced incidence of CAAs by 4–10% [1–7]. Persistent CAAs subsequently lead to thrombosis and stenotic lesions that result in myocardial ischemia and infarction [3] and aggressive management such as revascularization intervention is sometimes necessary for patients with these complications [1, 6, 7].

The American Heart Association (AHA) first released guidelines for the diagnosis and therapy of KD in 2004 (2) and most recently updated in 2017 [1]. The latest AHA guideline stratifies patients into five risk levels according to their relative risk of myocardial ischemia and infarction, with indications for subsets of each risk level using the status of the coronary artery and Z-score. Serial myocardial stress tests are recommended in addition to regular echocardiography in the presence of CAAs [1, 5, 6]. Long-term outcomes and reports of major adverse cardiac events (MACE) related to KD have been published [5, 8, 9]. A large nationwide survey in Japan between 1999 and 2010 found that 209 patients with giant CAAs had a 10-year survival rate of 94.3% and a total cardiac event-free rate of 0.68 [8]. A retrospective study of 500 CAAs in 2,860 KD patients in the US reported MACE in 24 patients (4.8%) [9].

In Thailand, the incidence of KD from 1998 to 2002 was reported to be between 2.14 and 3.43 cases per 100,000 children aged 0–5 years, and 15.6% of children had IVIG-resistant KD [10]. The reported prevalence of incomplete KD in children was 29% in a single-center study in Northern Thailand [11]. The long-term outcome of CAAs after treatment, however, has not been reported in the Thai population. We conducted a surveillance study to revisit the natural history of patients with CAAs after KD and to identify risk factors associated with MACE in the pediatric population in Thailand.

## Material and methods

Data were retrospectively collected from hospital databases from two large tertiary cardiac centers in Thailand: Faculty of Medicine Siriraj Hospital, Mahidol University, Bangkok and Faculty of Medicine Srinagarind Hospital, Khon Kaen University, Khon Kaen. The study was approved by the Siriraj Institutional Review Board, Faculty of Medicine, Siriraj Hospital, Mahidol University [Study number 294/2561 (EC1) and Srinagarind Hospital, Khon Kaen University (Reference No.HE611289). Both institutes waived informed consent from patients but required specific processes to ensure protection of subject confidentiality. All research methods were performed in accordance with Good Clinical Practice (GCP) guidelines and regulations.

Data from pediatric patients diagnosed with KD and CAAs between January 1, 1994 and June 30, 2019 were reviewed. Patients who had only a single echocardiography or angiography without any follow-up were considered as incomplete data and excluded. Demographic, clinical, initial laboratory, echocardiographic findings of CAAs and KD treatment were collected. Demographic data included gender, age at diagnosis of KD, clinical presentation, and criteria of diagnosis. Laboratory data recorded was erythrocyte sedimentation rate (ESR), white blood cell (WBC), and platelet count. The echocardiographic findings of CAAs were collected at the time of diagnosis, 6–8 weeks following diagnosis, and from the most recent follow-up visit. The dimensions of the right coronary artery (RCA), left main coronary artery (LMCA), left anterior descending artery (LAD), and left circumflex artery (LCx) were recorded with their

Z-scores. The maximal dimension of CAAs was used to categorize the patients with small, medium, or giant aneurysms. Based on the 2017 AHA criteria [1], CAAs were classified as small aneurysms (Z-score 2.5 to <5), medium aneurysms (Z-score 5 to <10), or large or giant aneurysms (Z-score 10 or an absolute dimension >8 mm). Treatment of KD included receiving IVIG at onset of fever (within 10 days or after 10 days), retreatment with IVIG, and receiving adjunctive anti-inflammation medications. Coronary angiographic findings and data of stress myocardial perfusion imaging (MPI) consisted of stress radionuclide imaging, stress cardiac cardiovascular magnetic resonance (CMR), and exercise stress test. Long-term outcomes were examined, including MACE and mortality using either the current hospital's databases or phone contact. MACE was defined as having cardiovascular-related illness including total coronary artery occlusion, heart failure, clinical or imaging evidence of myocardial ischemia (MI) by either stress CMR or stress radionuclide MPI, requirement of coronary artery bypass grafting (CABG), or percutaneous coronary intervention (PCI) following a diagnosis at their most recent follow-up visit in 2020.

## Statistical methods

Statistical analyses were performed using SPSS 20.0 for Windows (SPSS Inc., Chicago, IL, USA). Demographic, clinical, laboratory, cardiac imaging data, and KD treatment data were presented as a frequency with percentages for the categorical variables and mean ± SD or median with range for continuous variables. Factors associated with MACE were analyzed using univariate and multivariate analyses with cox proportional hazard models. The Kaplan-Meier method was used to analyze intervention-free rates for CABG or PCI in patients after diagnosis with KD. A statistically significant difference was considered with a $p$-value <0.05.

## Results

### Patient characteristics

A total of 658 pediatric patients were diagnosed with KD in the two cardiac centers between 1994 and 2019, and 170 of these children (65% male) who had CAAs following KD were included in the analysis (Fig 1). MACE was reported in 19 of 170 patients (11.2%), comprised of 12 CABG, 2 PCI, and 5 with evidence of myocardial ischemia and coronary occlusion.

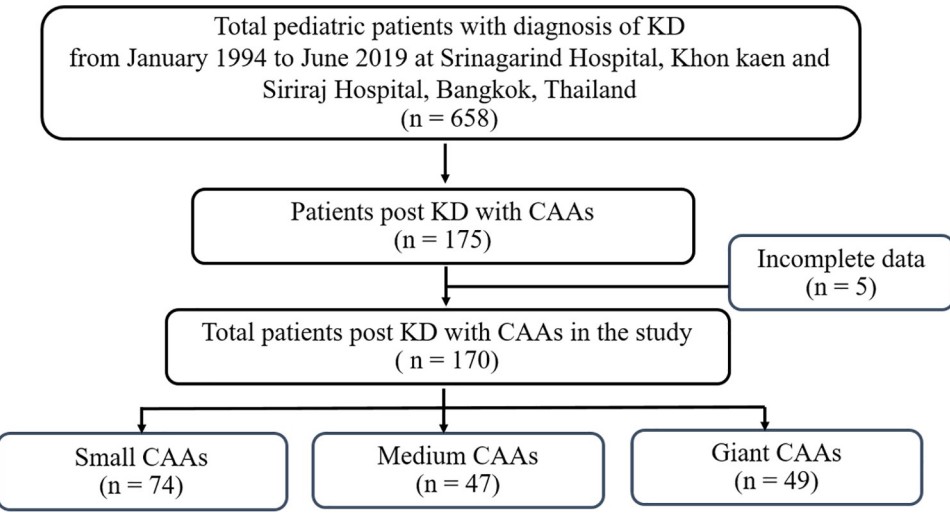

**Fig 1. Flow diagram of pediatric patients included in the analysis (n = 170).**

Demographic, clinical, and laboratory data, and KD treatments of the patients are shown in Table 1. The maximal Z-score of coronary dimensions at KD diagnosis was 6.3 (range 2.6 to 85.5). There were 89 patients who had CAAs that involved bilateral branches of coronary arteries. Sixteen children (9.4%) did not receive IVIG. Ninety children (52.9%) received timely IVIG treatment within 10 days of onset of fever. Retreatment of IVIG was reported for 25 (14.7%) of children. All patients received 80–100 mg/kg/day of aspirin during the acute phase of KD per standard of care in Thailand followed by standard low-dose aspirin (3–5 mg/kg/day). An additional anticoagulant, primarily warfarin, was administered with aspirin in 36

**Table 1. Baseline characteristics at initial diagnosis of Kawasaki disease.**

| Variable | Total (n = 170) | MACE (n = 19) | No MACE (n = 151) |
|---|---|---|---|
| Age at diagnosis (years) | 1.68 (0.2–12.5) | 2.72 (0.3–12.5) | 1.65 (0.2–9.9) |
| Site | | | |
| • Siriraj Hospital | 135 (79.4%) | 18 (94.7%) | 117 (77.5%) |
| • Srinagarind Hospital | 35 (20.6%) | 1 (5.3%) | 34 (22.5%) |
| Male sex | 111 (65.3%) | 17 (89.5%) | 94 (62.3%) |
| Typical KD | 66 (38.8%) | 7 (36.8%) | 59 (39.1%) |
| Atypical KD | 88 (51.8%) | 9 (47.4%) | 79 (52.3%) |
| Uncertain typical or atypical KD | 16 (9.4%) | 3 (15.8%) | 13 (8.6%) |
| Lack of IVIG treatment | 16 (9.4%) | 5 (26.3%) | 11 (7.3%) |
| Timing of IVIG treatment | | | |
| • ≤10 days of fever | 90 (52.9%) | 5 (26.3%) | 85 (56.3%) |
| • >10 days of fever | 51 (30%) | 8 (42.1%) | 43 (28.5%) |
| • Unknown timing | 13 (7.7%) | 1 (5.3%) | 12 (7.9%) |
| Onset of fever received IVIG (day) | 9 ± 4 | 12 ± 5 | 9 ± 4 |
| Retreatment with 2$^{nd}$ IVIG | 25 (14.7%) | 4 (21.1%) | 21 (13.9%) |
| Receiving adjunctive anti-inflammatory medication | 9 (5.3%) | 1 (5.3%) | 8 (5.3%) |
| WBC (/mm$^3$) | 17,851 ± 7,550 | 19,651 ± 7,466 | 17,732 ± 7,571 |
| Platelet (/mm$^3$) | 512,076 ± 189,334 | 652,250 ± 172,609 | 502,885 ± 187,376 |
| ESR (mm/hr) | 80 ± 29 | 93 ± 22 | 79 ± 29 |
| Degree of coronary artery | | | |
| • Small aneurysm | 74 (43.5%) | - | 74 (49%) |
| • Medium aneurysm | 47 (27.6%) | 2 (10.5%) | 45 (29.8%) |
| • Giant aneurysm | 49 (28.8%) | 17 (89.5%) | 32 (21.2%) |
| Initial Z-score of coronary dimension | 5.4 (0.2–85.5) | 23.2 (1.8–85.5) | 4.6 (0.2–34.9) |
| Maximal Z-score of coronary dimension | 6.3 (2.6–85.5) | 24.2 (6.7–85.5) | 5.4 (2.6–34.9) |
| Location of coronary artery aneurysms | | | |
| RCA | 32 (18.8%) | 1 (5.3%) | 31 (20.5%) |
| LAD+RCA | 32 (18.8%) | 7 (36.8%) | 25 (16.6%) |
| LMCA+LAD+RCA | 29 (17.1%) | 4 (21.0%) | 25 (16.6%) |
| LAD | 24 (14.1%) | - | 24 (15.9%) |
| LMCA | 16 (9.4%) | - | 16 (10.6%) |
| LAD+RCA+LCx | 4 (2.4%) | 3 (15.8%) | 1 (0.6%) |
| LMCA+LAD | 9 (5.3%) | 1 (5.3%) | 8 (5.3%) |
| LMCA+RCA | 21 (12.4%) | 2 (10.5%) | 19 (12.6%) |
| LMCA+LAD+RCA+LCx | 3 (1.7%) | 1 (5.3%) | 2 (1.3%) |

Data presented as n (%), mean ± SD and median (range)

MACE = major adverse cardiac event; KD = Kawasaki disease; IVIG = intravenous immunoglobulin; LMCA = left main coronary artery; LAD = left anterior descending artery; RCA = right coronary artery; LCx = left circumflex artery; WBC = white blood cell; ESR = erythrocyte sedimentation rate

**Table 2. Management of CAAs at different risk levels.**

|  | Small CAAs (n = 74) | Medium CAAs (n = 47) | Giant CAAs (n = 49) |
|---|---|---|---|
| Coronary angiography | 9 (12.2%) | 13 (27.7%) | 35 (71.4%) |
| Myocardial perfusion imaging | 5 (5.5%) | 13 (27.7%) | 1. (71.5%) |
| • Radionuclear stress MPI | 1 | 4 | 16 |
| • Stress CMR | 4 | 7 | 16 |
| • EST | - | 2 | 3 |
| Coronary interventions |  |  |  |
| • CABG | - | - | 12 (24.5%) |
| • PCI | - | 1 (2.1%) | 1 (2%) |
| • None | 74 (100%) | 46 (97.9%) | 36 (73.5%) |

Data is shown as n (%). CAAs = coronary artery aneurysms; MPI = myocardial perfusion imaging;

CMR = cardiovascular magnetic resonance; EST = exercise stress test; PCI = percutaneous coronary intervention;

CABG = coronary artery bypass grafting

patients (21.2%). Adjunctive anti-inflammatory medications were prescribed for 9 patients; 8 with steroids and 1 with abciximab (GP IIb/IIIa inhibitors).

Patients with CAAs were classified into three groups: small CAAs [74 patients (43.5%)], medium CAAs [47 patients (27.6%)], and giant CAAs [49 patients (28.8%)] (Fig 1). Table 2 illustrates the investigations and treatment of patients depending on the size of CAAs, which included coronary angiography, MPI, and coronary interventions. Coronary artery bypass grafting (CABG) was a mainstay treatment for KD with coronary occlusion and myocardial infarction. Fourteen patients (1 medium-sized CAA and 13 giant CAAs) required coronary intervention (8.2%). Importantly, 4 patients with giant CAAs required a second operation for CABG due to the re-stenosis of the coronary artery.

## Clinical outcomes and survival

Nineteen patients (11.2%) experienced MACE after a median (range) follow-up of 5.4 years (22 days to 23 years)]. Of 19 patients, 11 patients had clinically ischemic heart disease. Fourteen patients required coronary intervention, including 12 CABG and 2 PCI, a median of 4 years (0.01 to 9.5 years) after their diagnosis of KD. One patient who underwent CABG at 7 years-of-age had a car accident and died in 2019 at the age of 21 years. Five patients had evidence of chronic total occlusion of coronary artery with well-developed collateral circulation, no coronary intervention has yet been performed. Details of the 19 patients who experienced MACE are provided as S1 Table. Notably, no MACE reports were found for patients with small CAAs. The intervention-free survival of patients with giant CAAs was significantly lower than that in patients with small CAAs ($p$-value < 0.001) and medium CAAs ($p$-value = 0.001) (Fig 2). The intervention-free rate in patients with giant CAAs were 93.8%, 78.7%, and 52.2% at 1, 5, and 10 years, respectively. Furthermore, the cardiac event-free rate in these patients were 87.6%, 72.1%, and 50.7% at 1, 5, and 10 years, respectively (Fig 3). The progression and regression of CAAs are shown in Fig 4. Ninety-six patients (56.4%) had regression of CAAs to normal coronary artery. Of the small CAAs, 81% regressed to a normal size, and for medium aneurysms, 50% regressed to a normal size (Fig 4).

## Risk factors of MACE in children with CAAs following KD

Risk factors associated with MACE in patients with CAAs following KD are shown in Table 3. The absence of IVIG treatment, the presence of giant aneurysms and CAAs at bilateral

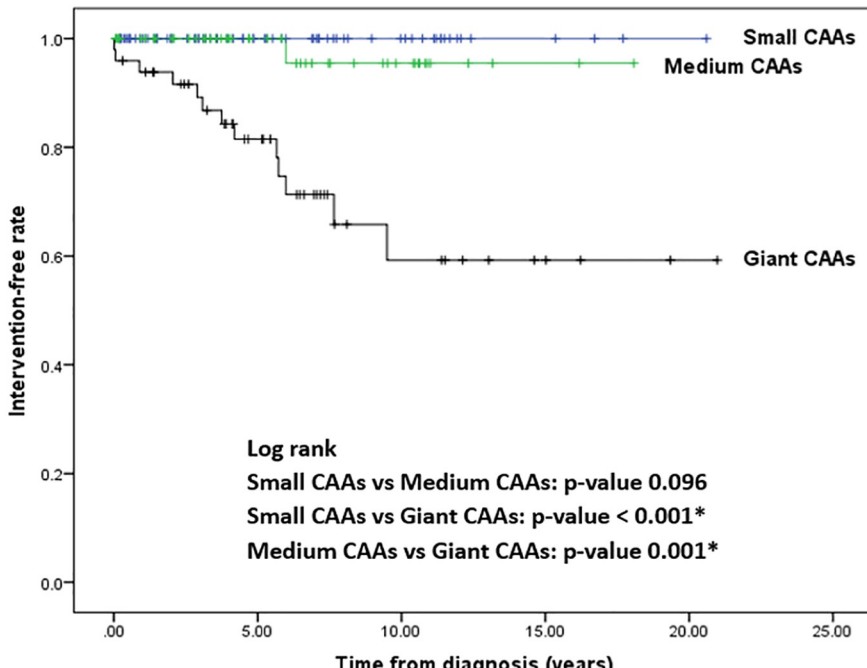

| Time | At Diagnosis | 1 year | 5 years | 10 years |
|---|---|---|---|---|
| Small CAAs | 74 | 61 | 33 | 16 |
| Medium CAAs | 47 | 41 | 24 | 11 |
| Giant CAAs | 49 | 45 | 27 | 9 |

**Fig 2. Kaplan-Meier estimates of the intervention-free rate of patients with Kawasaki disease who had coronary artery aneurysms (CAAs) (n = 170); small CAAs (n = 74; blue line), medium CAAs (n = 47; green line), and giant CAAs (n = 49; black line) from the time of initial diagnosis.**

branches of coronary involvement were identified as independent risks of MACE (adjusted HR 7.22; 95% CI 2.21–23.59; *p*-value = 0.001, adjusted HR 13.59; 95% CI 2.43–76.09; *p*-value = 0.003 and adjusted HR 6.19; 95% CI 1.24–30.92; *p*-value = 0.026, respectively) (Table 3).

## Discussion

This is the first study reporting the prevalence of MACE in children with KD and CAAs in Thailand. Retrospective analysis of data from two large referral hospitals over the last 25 years revealed that a quarter of children with KD had CAAs complications (n = 170). MACE was reported in 19 patients (11.2%); 14 of which required coronary interventions and 5 had evidence of coronary occlusion. Notably, no MACE was reported in children with small CAAs. The absence of IVIG treatment, the presence of giant aneurysms and bilateral branches of coronary involvement were identified as independent risk factors for MACE in these children. This study supports the consensus that close clinical monitoring for MACE following KD is critical, especially for children who did not receive IVIG or with medium to giant aneurysms.

The proportion of CAAs in KD using the 2017 AHA criteria in our study was 25.8%, comparable to the proportions of 24.6% in Japan [3] and 27.1% in USA [12]. Nevertheless, these

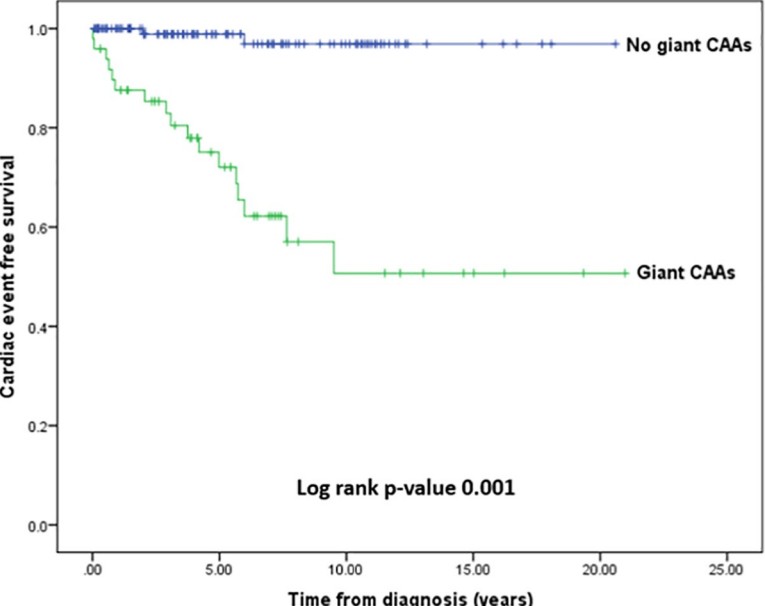

| Time | At Diagnosis | 1 year | 5 years | 10 years |
|------|------|------|------|------|
| No giant CAAs | 121 | 102 | 56 | 27 |
| Giant CAAs | 49 | 42 | 24 | 8 |

**Fig 3. Kaplan-Meier estimates of the cardiac event-free survival of patients with Kawasaki disease who had coronary artery aneurysms (CAAs) (n = 170); patients without giant CAAs (n = 121; blue line), and with giant CAAs (n = 49; green line) from time of initial KD diagnosis.**

values are higher than other studies where the prevalence of CAAs was found to vary between 3.6 and 17.4% [7, 13–15]. Data from a national survey on KD in Korea showed differences in the prevalence of CAAs using AHA criteria (21–42%) versus Japanese guideline criteria (18%) for the same sample population [16]. The use of different guidelines and Z-score formulas can lead to different CAA prevalence and diagnostic classification of KD. At our two centers; how-ever, the physicians routinely apply the AHA criteria and Z-scores, as used in this analysis. In addition, the lack of IVIG treatment (9.4%) and resistant IVIG KD (14.7%), which has been associated with the occurrence of coronary artery lesions in KD [13], were slightly higher, in this Korean cohort [16].

Regarding coronary dimensions over time, 56.4% of CAAs regressed to normal coronary artery dimensions in our study, with an overall regression in size of CAAs by 72%. This finding is consistent with a report by Friedman *et al.* that demonstrated a regression in CA aneurysms of 75% in KD patients [9]. Nevertheless, some patients still had a progressive increase in CAA sized over time, which indicates the need for caution and close follow-up [9, 13].

The incidence of MACE in our study in Thailand was 11.1%, which is higher than that the 4.8% reported in a US study in 2016 [9] but less than 21% reported in a study in Japan [17]. It has to be noted that MACE in our study included patients with clinical ischemic heart disease (6.4%) and asymptomatic patients with total coronary artery occlusion with imaging evidence

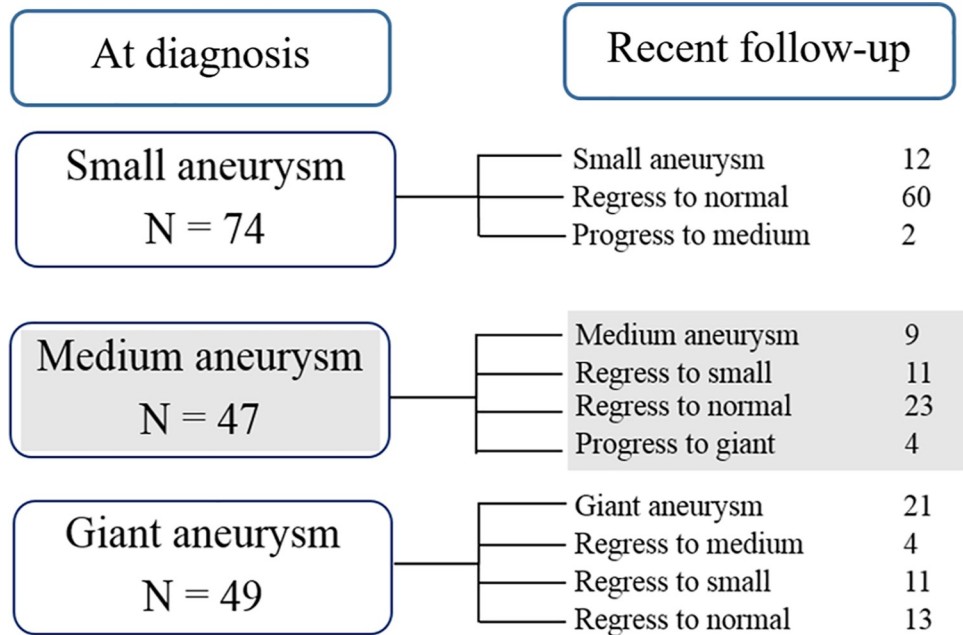

**Fig 4. Progression and regression of coronary artery aneurysms (CAAs) until the most recent follow-up visit (n = 170).**

of myocardial ischemia (MI) (4.7%). Our finding that most patients with MACE had giant aneurysms is consistent with previous publications [5, 9, 17]. Friedman and colleagues [9] reported no MACE in 313 patients with a coronary artery Z-score < 5, and 23% MACE in

**Table 3. Risk analysis of major adverse cardiac events (MACE) in Kawasaki disease with coronary aneurysms (CAAs).**

| Variable | Crude HR (95%CI) | *p*-value | Adjusted HR (95% CI) | *p*-value |
|---|---|---|---|---|
| Age at diagnosis <1 year | 0.71 (0.26–1.98) | 0.518 | | |
| Male sex | 4.27 (0.99–18.49) | 0.052 | 4.13 (0.9–18.87) | 0.068 |
| Atypical KD | 1.05 (0.39–2.82) | 0.924 | | |
| Lack of IVIG treatment | 4.72 (1.69–13.15) | 0.003* | 7.22 (2.21–23.59) | 0.001* |
| Retreatment with 2nd IVIG | 1.76 (0.58–5.33) | 0.315 | | |
| Received adjunctive anti-inflammatory medication | 0.95 (0.13–7.10) | 0.957 | | |
| Referral from other hospitals | 2.93 (0.68–12.67) | 0.151 | 0.51 (0.09–2.86) | 0.445 |
| Elevated ESR (mm/hr) | 1.01 (0.99–1.03) | 0.281 | | |
| Presence of giant CAAs | 20.6 (4.76–89.26) | <0.001* | 13.59 (2.43–76.09) | 0.003* |
| Maximal Z-score of coronary involvement | 1.14 (1.09–1.19) | <0.001* | | |
| Location of coronary artery aneurysm | | | | |
| -RCA | 0.28 (0.04–2.12) | 0.217 | | |
| -LAD | 0.39 (0–10.45) | 0.255 | | |
| -LAD+RCA | 2.42 (0.95–6.15) | 0.064 | | |
| Presence of CAAs in bilateral branches of coronary arteries | 8.09 (1.87–35.01) | 0.005* | 6.19 (1.24–30.92) | 0.026* |

Multivariate analysis by Cox regression

* Statistical significance at *p*-value < 0.05

MACE = major adverse cardiac event; KD = Kawasaki disease; IVIG = intravenous immunoglobulin; LMCA = left main coronary artery; LAD = left anterior descending artery; RCA = right coronary artery; WBC = white blood cell; ESR = erythrocyte sedimentation rate; CAAs = coronary artery aneurysms

patients with a coronary Z-score > 10. Similarly, the recent 34-institution international registry of 1,651 KD patients with CAAs [5] showed no MACE in patients with small CAAs, while the patients with giant aneurysms had a cumulative incidence of significant luminal narrowing (20±3%), coronary artery thrombosis (18±2%), and composite MACE at 10 years (14±2%).

In the present study, 14 patients underwent coronary interventions after KD diagnosis 0.01 to 9.5 years. These include our anecdotal five-case series institutionally published in 2006 [18]. Overall, our 10-year cardiac event-free rate in patients with giant CAAs was 50.7%, which is less than the 65–75% reported in prior studies [8, 17, 19]. The independent risk factors of MACE in KD with CAAs in our study were the presence of giant aneurysms, bilateral branches of coronary involvement and a lack of IVIG treatment. Aneurysm size, a higher CAA Z-score, and a greater number of coronary artery branches being affected in KD patients have been shown to be important factors risks of MACE [5, 8, 9, 17, 19]. Aside from the larger size of CAAs, age less than 60 months, recurrent KD, parental history of KD, delayed admission, and IVIG-resistant KD have also been reported to lead to worse coronary outcomes at >30 days following diagnosis [13], while hypoalbuminemia was an additional risk of progressive coronary dilatation at one-year post KD diagnosis [20].

## Study limitations

Our study has several limitations. Firstly, due to retrospective nature, there will be selection bias. To reduce the bias, we made every effort to only include patients with KD who had evidence of CAAs by echocardiographic studies and on follow-up at either of the centers. Echocardiographic data and clinical outcomes at their most recent follow-up in October 2020 were recorded, and patients who had a recent follow-up prior to 2020 were contacted via phone to assess their MACE and clinical status. Eighty-seven (51.2%) patients had their recent clinical status assessed in October 2020. Of the 83 patients with no record in 2020, their recent status based on the available medical database. Another limitation was the minimal variability for the clinical investigations, especially the type of cardiac stress test and adjunctive anti-inflammation treatment used at the two centers. Future research may further assess KD patients without CAAs regarding their coronary changes and MACE rate. In addition, change in diastolic properties and left atrial reservoir function may be interesting to research in longitudinal cohort of patients with KD [21].

## Conclusion

The MACE rate was 11.2% for children KD with CAAs. A lack of IVIG treatment, the presence of giant aneurysms and bilateral branches of coronary involvement were independent risk factors for MACE in our population. Patients with giant CAAs following KD had an intervention-free survival rate that was significantly lower than that for patients with small and medium CAAs. The cardiac event and intervention-free rate at 10 years in patients with giant CAAs was ~50%. Once medium to giant aneurysms develop, cautious surveillance and early recognition of cardiac complications is recommended.

## Supporting information

**S1 Checklist.**
(PDF)

**S1 Table. Characteristics of 19 patients in the cohort who had major adverse cardiac events (MACE).**
(DOCX)

**S1 File.**
(ZIP)

## Acknowledgments

The authors acknowledge Siriraj Institute of Clinical Research (SICRES) and Dr. Tim R. Cressey (AMS-PHPT Research Unit, Faculty of Associated Medical Sciences, Chiang Mai University) for his support in the manuscript development. The authors also thank Dr. Julaporn Pooliam, Clinical Epidemiology Unit, Office of Research and Development, Faculty of Medicine, Siriaj Hospital for her assistance with the statistical analysis in addition to Dr. Glen Wheeler for his proofreading and editing.

## Author Contributions

**Conceptualization:** Kanokvalee Santimahakullert, Chodchanok Vijarnsorn.

**Data curation:** Chodchanok Vijarnsorn.

**Formal analysis:** Kanokvalee Santimahakullert, Chodchanok Vijarnsorn.

**Funding acquisition:** Chodchanok Vijarnsorn.

**Investigation:** Kanokvalee Santimahakullert, Chodchanok Vijarnsorn, Sappaya Khrongsrattha.

**Methodology:** Kanokvalee Santimahakullert, Chodchanok Vijarnsorn.

**Project administration:** Chodchanok Vijarnsorn.

**Resources:** Chodchanok Vijarnsorn, Yuttapong Wongswadiwat, Prakul Chanthong, Manat Panamonta, Paradorn Chan-on, Kritvikrom Durongpisitkul, Paweena Chungsomprasong, Supaluck Kanjanauthai, Jarupim Soongswang.

**Supervision:** Chodchanok Vijarnsorn.

**Validation:** Chodchanok Vijarnsorn.

**Visualization:** Kanokvalee Santimahakullert, Chodchanok Vijarnsorn.

**Writing – original draft:** Kanokvalee Santimahakullert, Chodchanok Vijarnsorn.

**Writing – review & editing:** Kanokvalee Santimahakullert, Chodchanok Vijarnsorn, Yuttapong Wongswadiwat, Prakul Chanthong, Sappaya Khrongsrattha, Manat Panamonta, Paradorn Chan-on, Kritvikrom Durongpisitkul, Paweena Chungsomprasong, Supaluck Kanjanauthai, Jarupim Soongswang.

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
