## [Decision Letter · Decision Letter 0]

9 Nov 2021

PONE-D-21-17324

A Retrospective Cohort Study of Major Adverse Cardiac Events in Children Affected by Kawasaki Disease with Coronary Artery Aneurysms in Thailand

PLOS ONE

Dear Dr. Vijarnsorn,

Thank you for submitting your manuscript to PLOS ONE. After careful consideration, we feel that it has merit but does not fully meet PLOS ONE’s publication criteria as it currently stands. Therefore, we invite you to submit a revised version of the manuscript that addresses the points raised during the review process.

We look forward to receiving your revised manuscript.

Kind regards,

Dong Keon Yon, MD, FACAAI

Academic Editor

PLOS ONE

Additional Editor Comments (if provided):

Many thanks for your consideration to submit in Plos One. The reviewers and I read it with great interest, please address adequately comments of the reviewers.

#1. A Retrospective Cohort Study of Major Adverse Cardiac Events in Children Affected by Kawasaki Disease with Coronary Artery Aneurysms

-> Major Adverse Cardiac Events in Children Affected by Kawasaki Disease with Coronary Artery Aneurysms: a two-center, retrospective cohort study in Thailand

#2. Please discuss the paper below.

Kang SJ, Kwon YW, Hwang SJ, Kim HJ, Jin BK, Yon DK. Clinical Utility of Left Atrial Strain in Children in the Acute Phase of Kawasaki Disease. J Am Soc Echocardiogr. 2018 Mar;31(3):323-332. doi: 10.1016/j.echo.2017.11.012. Epub 2018 Jan 3. PMID: 29305035.

Journal Requirements:

 "No financial disclosures or outside funding were present. The funders had no role in study design, data collection, analysis, decision to publish, or preparation of the manuscript." 

Reviewers' comments:

Reviewer's Responses to Questions

**Comments to the Author**

1. Is the manuscript technically sound, and do the data support the conclusions?

Reviewer #1: Yes

Reviewer #2: Yes

2. Has the statistical analysis been performed appropriately and rigorously? 

Reviewer #1: Yes

Reviewer #2: Yes

3. Have the authors made all data underlying the findings in their manuscript fully available?

Reviewer #1: Yes

Reviewer #2: Yes

4. Is the manuscript presented in an intelligible fashion and written in standard English?

Reviewer #1: Yes

Reviewer #2: Yes

5. Review Comments to the Author

Reviewer #1: Santimahakullert and colleagues present intruiguin data on a relatively large retrospective cohort of KD patients undergone treatment in Thailand. Specifics of clinical care are likely related to geographical area dn healthcare logistics, but these data provide confirming data on a number of clinical risk factors, with supporting long-term follow-up that is a plus of this study design.

I would like to raise some points to be considered before consideration for publication can be given.

- Time between symptom onset and diagnosis should be provided and treated as risk factor in the Cox regression model

- MACE: It is unclear if routine and universal myocardial perfusion study or stress test was implemented in the longitudinal follow-up of this cohort. Accordingly, it is of concern the inclusion of physician-driven ascertainment of some component of the outcome. Data should be reanalyzed taking out clinical or imaging evidence of myocardial ischemia (MI) by either stress CMR or stress radionuclide MPI component of the composite endpoint

-Comparison between KD patients with and without CAAs would be informative

-Low use of IV immunoglobulin should be further defined and explained

-Decision-making regarding need for additional myocardial ischemia testing and/or invasive angiography with comparative tables should be provided

- Missing data and eventual handling of missing data should be specified

Reviewer #2: The authors provide an instructional review of long term cardiac complication from KD in Thailand. Their results are a valuable contribution to the literature on this subject. They identify several risk factors that numerous other studies have shown, including the absence of IVIG treatment, presence of giant coronary aneurysms, location at branch points. Therefore, their results do not provide any surprising findings other than 2 pieces of data. First, the incidence of adverse events (called "MACE" in the manuscript) was much higher in Thailand than in the US, 11.1 vs 4.8%. This is quite a discrepancy and should be addressed with at least some speculation on this finding, or a closer look at the data to find an explanation. Second, the authors found that a surprisingly high proportion (28 of 49, 57%) of regression of giant aneurysms. Since giant aneurysms are the key risk factor for late complications, it would be helpful to make some comments on this particular finding.

6. PLOS authors have the option to publish the peer review history of their article (what does this mean?). If published, this will include your full peer review and any attached files.

Reviewer #1: No

Reviewer #2: **Yes: **Lee Beerman MD

---

## [Author Response · Author response to Decision Letter 0]

18 Dec 2021

December 5, 2021

Dr. Dong Keon Yon, MD, FACAAI

Academic Editor

PLOS ONE

Manuscript Reference Number: PONE-D-21-17324

Title: A Retrospective Cohort Study of Major Adverse Cardiac Events in Children Affected by Kawasaki Disease with Coronary Artery Aneurysms in Thailand

Dear Editor,

Thank you for your careful consideration of our manuscript and we appreciate the extensive review. Please find below our responses (in blue font) to each of the points raised:

Additional Editor Comments (if provided):

Many thanks for your consideration to submit in Plos One. The reviewers and I read it with great interest, please address adequately comments of the reviewers.

#1. A Retrospective Cohort Study of Major Adverse Cardiac Events in Children Affected by Kawasaki Disease with Coronary Artery Aneurysms

-> Major Adverse Cardiac Events in Children Affected by Kawasaki Disease with Coronary Artery Aneurysms: a two-center, retrospective cohort study in Thailand

Answer: Sorry for the confusion. To clarify, we submitted the above referenced paper to the preprint research square platform in March 2021. Please note that this preprint version had not been published in any journal at that time. We subsequently reviewed and reanalyzed our data and new results were added before submitting the current version of the manuscript entitles “A Retrospective Cohort Study of Major Adverse Cardiac Events in Children Affected by Kawasaki Disease with Coronary Artery Aneurysms” to PLOS ONE in title on May 30, 2021. 

#2. Please discuss the paper below.

Kang SJ, Kwon YW, Hwang SJ, Kim HJ, Jin BK, Yon DK. Clinical Utility of Left Atrial Strain in Children in the Acute Phase of Kawasaki Disease. J Am Soc Echocardiogr. 2018 Mar;31(3):323-332. doi: 10.1016/j.echo.2017.11.012. Epub 2018 Jan 3. PMID: 29305035.

Answer: Thank you pointing out this interesting and well-designed study. The peak left atrial longitudinal strain (PALS) and diastolic property were explored in acute phase and convalescent phase of KD patients compared to an age-matched control group. This article found that impaired left atrial reservoir function could be detected as decreased PALS, LV longitudinal peak systolic strain rate in patients with acute phase of KD. Unfortunately, given the retrospective nature of our study we were unable to explore these parameters; nevertheless, we have noted this information and cited this paper in the revised manuscript.

Journal Requirements:

Answer: We have formatted the manuscript to meet PLOS ONE's style requirements and the references have been revised to Vancouver style per template.

"No financial disclosures or outside funding were present. The funders had no role in study design, data collection, analysis, decision to publish, or preparation of the manuscript." 

Answer: There was no external grant or internal funding received for this study. This study involved reviewing hospital-databases following IRB approval. No specific funding was sought for this study. 

Answer: As no external grant or internal funding was received we have stated “No financial disclosures were present and no external or internal funding was received.”

Answer: There was no external grant or internal funding received for this study. None of the authors received specific funding for this work.

Answer: We stated that “The authors received no specific funding for this work.”

Answer: The datasets generated and/or analyzed are not publicly available as they contain sensitive information that could potentially compromise patient confidentiality. However, the data could be shared on request from the corresponding author and ethics committee (contact: siriraj_irb@mahidol.ac.th).

b) If there are no restrictions, please upload the minimal anonymized data set necessary to replicate your study findings as either Supporting Information files or to a stable, public repository and provide us with the relevant URLs, DOIs, or accession numbers. For a list of acceptable repositories, please see http://journals.plos.org/plosone/s/data-availability#loc-recommended-repositories. We will update your Data Availability statement on your behalf to reflect the information you provide.

Answer: Please kindly refer to answer 3.a.

Answer: Thank you for the information.

Answer: In-text citation for Table S1 has been included in the results: Clinical Outcomes and Survival.

Reviewers' comments:

Comments to the Author

1. Is the manuscript technically sound, and do the data support the conclusions?

Reviewer #1: Yes

Reviewer #2: Yes

Answer: Thank you for your consideration.

2. Has the statistical analysis been performed appropriately and rigorously?

Reviewer #1: Yes

Reviewer #2: Yes

Answer: Thank you for your consideration.

3. Have the authors made all data underlying the findings in their manuscript fully available?

Reviewer #1: Yes

Reviewer #2: Yes

Answer: Thank you for your consideration.

4. Is the manuscript presented in an intelligible fashion and written in standard English?

Reviewer #1: Yes

Reviewer #2: Yes

Answer: Thank you for your consideration.

5. Review Comments to the Author

Reviewer #1: Santimahakullert and colleagues present intriguing data on a relatively large retrospective cohort of KD patients undergone treatment in Thailand. Specifics of clinical care are likely related to geographical area and healthcare logistics, but these data provide confirming data on a number of clinical risk factors, with supporting long-term follow-up that is a plus of this study design.

I would like to raise some points to be considered before consideration for publication can be given.

- Time between symptom onset and diagnosis should be provided and treated as risk factor in the Cox regression model

Answer: Median time between symptom onset, such as fever, to diagnosis of Kawasaki disease was 7 days (range 0 to 36 days). Of note, there were 29 patients with late presentation with no clear history of fever leading to some uncertainty of the time between symptom onset and diagnosis. Consequently, delayed time between symptom onset and diagnosis was not related with MACE (p = 0.693). As patients with delayed diagnosis often received IVIG after 10 days of symptom onset or do not receive IVIG treatment, the authors used lack of IVIG treatment as one of the risk factors in the analysis. 

- MACE: It is unclear if routine and universal myocardial perfusion study or stress test was implemented in the longitudinal follow-up of this cohort. Accordingly, it is of concern the inclusion of physician-driven ascertainment of some component of the outcome. Data should be reanalyzed taking out clinical or imaging evidence of myocardial ischemia (MI) by either stress CMR or stress radionuclide MPI component of the composite endpoint

Answer: Thank you for your question. As noted, a major adverse cardiac event (MACE) in our study was defined as having cardiovascular-related illness that included total coronary artery occlusion, heart failure, clinical or imaging evidence of myocardial ischemia (MI) by either stress cardiovascular magnetic resonance or stress radionuclide MPI, requirement of coronary artery bypass grafting (CABG), or percutaneous coronary intervention (PCI) following a diagnosis at their most recent follow-up visit in 2020, which may lead physician-driven ascertainment of some component of the outcome. Of 19 patients with MACE, 11 patients (6.4%) had clinical symptoms and 8 patients (4.7%) had evidence of perfusion deficit leading coronary intervention. The data was reanalyzed removing imaging evidence of myocardial ischemia (MI) by either stress CMR or stress radionuclide MPI following your suggestion. The HR using data of clinical MACE (n=11) are shown in the table below.

Table: Risk analysis of clinical major adverse cardiac event in Kawasaki disease with coronary aneurysms (CAAs)

Variable Crude HR (95%CI) p-value Adjusted HR

(95% CI) p-value

Age at diagnosis <1 year 0.44 (0.1-2.05) 0.296 

Male sex 4.85 (0.62-37.94) 0.132# 4.14 (0.53-32.51) 0.177

Atypical KD 0.81 (0.23-2.89) 0.814 

Lack of IVIG treatment 6.52 (1.9-22.29) 0.003# 9.37 (2.68-32.79) <0.001*

Retreatment with 2nd IVIG 1.45 (0.31-6.72) 0.636 

Received adjunctive anti-inflammatory medication 0.05 (0-2483) 0.574 

Referral from other hospitals 1.46 (0.31-6.74) 0.632 

Elevated ESR (mm/hr) 22.79 

(0-18160880) 0.652 

Presence of giant CAAs 218.05 

(0.63-75218.81) 0.071# N/A N/A

Presence of CAAs in bilateral branches of coronary arteries 9.043 (1.16-70.65) 0.036#

 12.36 (1.54-98.9) 0.018*

Multivariate analysis by Cox regression

# Statistical significance at p-value < 0.2

* Statistical significance at p-value < 0.05

KD=Kawasaki disease; IVIG=intravenous immunoglobulin; ESR=erythrocyte sedimentation rate; CAAs=coronary artery aneurysms; N/A=not applicable

Using 11 patients with clinical MACE as the endpoint, lack of IVIG, presence of giant coronary aneurysms and bilateral coronary involvement were identified as dependent risk factors of MACE (p < 0.2) on univariate analysis. The absence of IVIG treatment and CAAs at bilateral branches of coronary involvement were identified as independent risks of MACE. All 11 patients with clinical MACE had giant aneurysms but this risk factor was unable to be assessed in the multivariate analysis even though it is likely an important risk factor of MACE. These results are consistent with the risk factors we identified in the original analysis. Nevertheless, we should be cautious of using only clinical MACE as some children cannot report their ischemic heart symptoms and this leads to late detection. Recent KD guidelines suggest to perform serial myocardial stress tests and manage the patients who are at risk. In our setting, when we detect perfusion deficit on myocardial stress tests, CAG would be performed and treatment would be considered. We therefore decided to include the eight patients who had coronary occlusion and perfusion deficit as patients with MACE. 

We have added the number of patients with clinical MACE to the result and discussion section as suggested.

-Comparison between KD patients with and without CAAs would be informative

Answer: The inclusion criteria of this retrospective study only included consecutive patients with KD who had CAAs. Patients with no CAAs were not include in this analysis. Interestingly, no patients with small CAAs experienced MACE which may indicate that moderate and giant CAA could be risky more than small CAAs. Our team is currently performing research in KD patients without CAAs and assessing coronary changes and MACE and we expect the results will help answer this question. 

-Low use of IV immunoglobulin should be further defined and explained

Answer: The use of IVIG in Thailand between 1990-2004 was limited due to lack of availability in the national health program and late diagnosis. IVIG usage in our retrospective study (between 1994 and 2019) was 91%. 

-Decision-making regarding need for additional myocardial ischemia testing and/or invasive angiography with comparative tables should be provided

Answer: The decision making of additional myocardial ischemia testing and/or invasive angiography is based on the 2017 AHA KD guideline (McCrindle BW, Rowley AH, Newburger JW, Burns JC, Bolger AF, Gewitz M, et al. Diagnosis, Treatment, and Long-Term Management of Kawasaki Disease: A Scientific Statement for Health Professionals From the American Heart Association. Circulation. 2017;135(17):e927-e99), especially for patients with moderate to giant CAAs. The retrospective nature of our analysis will inevitably include some bias. The patients diagnosed with KD prior to 2017 would be performed additional testing as 2004 guideline suggestion. Traditional tests included coronary angiography and nuclear MPI while the current tests included stress cardiovascular magnetic resonance, exercise stress test up to patients’ classification. 

- Missing data and eventual handling of missing data should be specified

Answer: We have highlighted this point as a study limitation. Clinical outcomes and echocardiographic data at the patients’ most recent follow-up in October 2020 were recorded. Patients who had a recent follow-up prior to 2020 were contacted via phone to assess clinical status and any MACE. Eighty-seven (51.2%) patients had their recent clinical status assessed in October 2020. Of the 83 patients with no record in 2020, their recent status based on the available medical database was used for data analysis.

Reviewer #2: The authors provide an instructional review of long term cardiac complication from KD in Thailand. Their results are a valuable contribution to the literature on this subject. They identify several risk factors that numerous other studies have shown, including the absence of IVIG treatment, presence of giant coronary aneurysms, location at branch points. Therefore, their results do not provide any surprising findings other than 2 pieces of data. 

-First, the incidence of adverse events (called "MACE" in the manuscript) was much higher in Thailand than in the US, 11.1 vs 4.8%. This is quite a discrepancy and should be addressed with at least some speculation on this finding, or a closer look at the data to find an explanation. 

Answer: The incidence of MACE in our study in Thailand was 11.1%, which is higher than the 4.8% reported in a US study in 2016, but less than the 21% reported in a study in Japan in 2017. One explanation may be the ethnicity of the populations. Notably, our study included 8 patients who were asymptomatic but had evidence of chronic total occlusion of coronary artery in the MACE group. If we exclude these patients, the proportion of symptomatic MACE patients was 11/170 (6.4%). However, we decided to include these patients with MACE as the study population was children and likely unable to report the ischemic symptoms. The myocardial stress tests may aid us to detect myocardial ischemia prior to symptom onset. 

-Second, the authors found that a surprisingly high proportion (28 of 49, 57%) of regression of giant aneurysms. Since giant aneurysms are the key risk factor for late complications, it would be helpful to make some comments on this particular finding.

Answer: Of 49 patients with giant coronary aneurysms (CAAs), 21 patients (43%) had persistent giant CAAs, 4 patients (8%) regressed to medium sized CAAs, 11 patients (22.5%) regressed to small CAAs and 13 patients (26.5%) regressed to normal size. Initial evaluation of giant CAAs was performed by echocardiography. The progression or regression of CAAs were subsequently assessed by echocardiography and coronary angiography/ computerized tomography/ coronary magnetic resonance angiography. Associated factors of regression of giant CAAs were explored using univariate analysis and the results are shown in the table below. In this subgroup analysis, age of less than 1 year at the onset of Kawasaki syndrome, female sex, typical KD, received IVIG treatment, retreatment with 2nd IVIG, presence of CAAs in unilateral branch of coronary arteries were not associated with favorable resolution of CAAs. A larger study population is needed to provide robust predictors of giant CAAs regression. Please note that the morphology of CAAs such as fusiform CAAs or location of CAAs (distal or proximal portion of coronary artery) found to be favorable factors in a previous publication(1) were not assessed in our study due to lack of data. A study of 120 patients with CAAs in Eastern China(2) showed patients aged ≤ 1 year, received initial intravenous immunoglobulin (IVIG) treatment after the 10th day of illness, and IVIG non-responders were associated with the regression of persistent CAAs.

1) Takahashi M, Mason W, Lewis AB. Regression of coronary aneurysms in patients with Kawasaki syndrome. Circulation. 1987;75(2):387-94. 

2) Tang Y, Yan W, Sun L, Xu Q, Ding Y, Lv H. Coronary artery aneurysm regression after Kawasaki disease and associated risk factors: a 3-year follow-up study in East China. Clin Rheumatol. 2018;37(7):1945-51.

Table: Univariate analysis of factors associated with regression of giant coronary aneurysms (n=49)

Variable Crude OR (95%CI) p-value

Age at diagnosis < 1 year 1.67 (0.76-3.66) 0.201

Female sex 1.05 (0.45-2.49) 0.905

Typical KD 0.50 (0.22-1.13) 0.094

Received IVIG treatment 1.11 (0.39-3.29) 0.842

Retreatment with 2nd IVIG 0.95 (0.36-2.54) 0.925

Not received adjunctive anti-inflammatory medication 21.71 (0.004-1117007) 0.483

Presence of CAAs in unilateral branch of coronary arteries 1.79 (0.75-4.26) 0.188

Received IVIG after 10 days 1.28 (0.54-3.02) 0.573

KD=Kawasaki disease; IVIG=intravenous immunoglobulin; CAAs=coronary artery aneurysms

Best wishes,

Dr. Chodchanok Vijarnsorn

Division of Pediatric Cardiology

Department of Pediatrics, Faculty of Medicine

Siriraj Hospital, Mahidol University THAILAND

---

## [Decision Letter · Decision Letter 1]

12 Jan 2022

A Retrospective Cohort Study of Major Adverse Cardiac Events in Children Affected by Kawasaki Disease with Coronary Artery Aneurysms in Thailand

PONE-D-21-17324R1

Dear Dr. Vijarnsorn,

We’re pleased to inform you that your manuscript has been judged scientifically suitable for publication and will be formally accepted for publication once it meets all outstanding technical requirements.

Kind regards,

Dong Keon Yon, MD, FACAAI

Academic Editor

PLOS ONE

Additional Editor Comments (optional):

I congratulate you on this mesmerizing paper.

Reviewers' comments:

Reviewer's Responses to Questions

**Comments to the Author**

1. If the authors have adequately addressed your comments raised in a previous round of review and you feel that this manuscript is now acceptable for publication, you may indicate that here to bypass the “Comments to the Author” section, enter your conflict of interest statement in the “Confidential to Editor” section, and submit your "Accept" recommendation.

Reviewer #2: All comments have been addressed

2. Is the manuscript technically sound, and do the data support the conclusions?

Reviewer #2: Yes

3. Has the statistical analysis been performed appropriately and rigorously? 

Reviewer #2: Yes

4. Have the authors made all data underlying the findings in their manuscript fully available?

Reviewer #2: Yes

5. Is the manuscript presented in an intelligible fashion and written in standard English?

Reviewer #2: Yes

6. Review Comments to the Author

Reviewer #2: The authors have effectively answered the questions and suggestions of the reviewers and I would now favor acceptance.

7. PLOS authors have the option to publish the peer review history of their article (what does this mean?). If published, this will include your full peer review and any attached files.

Reviewer #2: No

---

## [Editor Report · Acceptance letter]

17 Jan 2022

PONE-D-21-17324R1 

A Retrospective Cohort Study of Major Adverse Cardiac Events in Children Affected by Kawasaki Disease with Coronary Artery Aneurysms in Thailand 

Dear Dr. Vijarnsorn:

I'm pleased to inform you that your manuscript has been deemed suitable for publication in PLOS ONE. Congratulations! Your manuscript is now with our production department. 

Kind regards, 

on behalf of

Dr. Dong Keon Yon 

Academic Editor

PLOS ONE